# Research on Integrated Scheduling of Multi-Mode Emergency Rescue for Flooding in Chemical Parks

Bowen Guo [1] and Wei Zhan [2,*]

1   School of Engineering Science, University of Chinese Academy of Sciences, Beijing 100049, China
2   School of Emergency Management Science and Engineering, University of Chinese Academy of Sciences, Beijing 100049, China
*   Correspondence: weizhan@ucas.ac.cn

**Abstract:** As the scale of the chemical park industry continues to expand, the impact of flooding on the park's people and surrounding environment increases. This paper uses project scheduling theory to optimize the emergency rescue process in order to alleviate the suffering of affected people, promote the sustainable development of society and the environment, and take into account the characteristics of the dynamic evolution of flooding in chemical parks and the periodic renewal of emergency resources. We constructed a proactive–reactive multi-mode emergency rescue integrated scheduling model that aims to minimize the loss of affected people in the early stage of flooding and minimize the sum of the total deviation of the start time and end time of activities before and after reactive scheduling in the later stages of flooding. Furthermore, an ant colony algorithm was designed to solve the constructed model. Next, the effectiveness of the proposed model and solution algorithm was verified using simulations of actual cases. The calculation results show that using proactive–reactive integrated scheduling can improve the efficiency of emergency rescue and reduce the loss of affected people. Moreover, if a multi-mode rescue strategy is adopted, emergency rescue scheduling under four different resource combinations can reduce rescue duration and loss of affected people. The model can provide a decision reference for sustainable emergency rescue scheduling in chemical parks during a flood.

**Keywords:** flood disaster emergency management; chemical parks; emergency rescue scheduling; sustainable development; ant colony algorithm

## 1. Introduction

In recent years, given the abnormal climate caused by global warming, flooding has occurred frequently, which significantly affects the normal operation order of chemical parks [1]. As essential engines that drive local economic development, the cluster development process of chemical parks is accelerating, the scale of the industry is expanding, and the degree of gathering of hazardous sources is increasing. This development trend leads to the increased probability of a domino effect of disaster accidents in chemical parks under the influence of floods [2,3]. The overall risk of hazardous chemical accidents is also rising [4]. This has caused significant damage to the safety of life and property in parks and has severely impacted the local ecological environment and sustainable economic development.

Floods and their secondary hazards acting on chemical parks can cause serious hazards to heavy multi-factors such as production and storage units, utility infrastructure, and lifeline systems in the parks [5]. In particular, these hazards can cause flooding failure, building collapse, material leakage, as well as damage to equipment and facilities in production and storage units, and public infrastructure. They can also lead to other hazards, which can lead to poisoning, combustion, explosions, and other accidents. The lifeline system can lead to trapped personnel, communication interruption, and blocked emergency

rescue. The accident chain caused by flooding may cause property damage, threaten the life and safety of park personnel and nearby residents, and cause environmental pollution, which will have a serious negative impact on the sustainable development of society [6]. Sudden flooding and the derivative disasters suffered by a chemical park are a complex system with non-linear evolutionary characteristics and connections between their many constituent units, which also leads to substantial uncertainty when carrying out emergency rescue work during flooding. Emergency rescue work involves numerous subjects, and there is tandem and parallel synchronization between various rescue activities [7]. To minimize the casualties and environmental damage caused by disasters in the park and promote sustainable development, it is necessary to accurately draw the disaster evolution scenario and control the disaster evolution law based on disaster data information within the shortest time after the occurrence of the disaster, in order to understand the dynamic situation of rescue needs and provide a reasonable and effective emergency rescue scheduling plan [8]. An unreasonable emergency rescue strategy will delay the golden rescue time and may even lead to severe consequences, such as loss of affected people and property, waste of resources, and environmental damage.

Emergency rescue is a vital part of the emergency response phase of a disaster. Before taking rescue actions, a scientific and effective assessment of the safety risks, accident hazards, and emergency response capabilities of chemical parks is needed to ensure the sustainability of rescue efforts. Some scholars have recently conducted research on related issues. For example, Zeng et al. [9] combined the vulnerability model, probability estimation, and other methods to quantify a risk analysis for a domino effect caused by a flood in a chemical industrial park and proposed risk prevention measures to make the park more resilient and safe when encountering flood disasters. Zhang et al. [10] combined Bayesian theory to construct an oil pipeline accident probability analysis model for assessing the magnitude of each accident risk to facilitate the formulation of corresponding emergency response plans. Lan et al. [11] introduced network analysis techniques to assess the domino effect associated with NaTech in process clusters in chemical parks, which provided a reasonable basis for implementing an emergency response. Reniers et al. [12] used mathematical networks to demonstrate that the disaster resilience of chemical parks obeys a power–law distribution and also illustrated the attenuation-based safety of chemical parks with examples. Liu et al. [13] proposed a PCA-BP neural network-based emergency rescue capability assessment model to evaluate the professional capability of chemical professional emergency rescue teams. Wang et al. [14] constructed an evaluation model for the emergency response capacity of environmental pollution accidents with a small town as the research object.

The abovementioned studies provide a certain reference basis for carrying out emergency response work for disaster accidents in chemical parks. In the emergency response phase, different scholars have conducted studies on different rescue efforts. Some scholars have conducted research on the problem of emergency path planning in chemical parks. For example, Xu et al. [15] constructed a multi-indicator emergency risk assessment and emergency route-planning model by considering chemical accident scenarios and emergency behavior characteristics of different individuals. Phark et al. [16] used machine learning algorithms to study the decision model for the emergency evacuation of chemical accidents. In addition, some scholars have studied the problem of emergency resource scheduling. Li et al. [17] developed a multi-objective emergency material scheduling optimization model with time window constraints for offshore oil spill accidents. To address the problem of resource scarcity in the emergency rescue sector, Luscombe et al. [18] constructed a dynamic resource scheduling model, which effectively improved the efficiency of medical resource matching. Liu et al. [19] considered the dynamic changes in supply and demand, and constructed an emergency material scheduling model to minimize the damage caused by chemical spills to rivers. In addition, Liu et al. [20] also designed an emergency material scheduling model for chemical spills that considered both the timeliness and cost-effectiveness of emergency relief. Some other scholars have conducted

in-depth studies on the location of humanitarian response facilities. Lu et al. [21] developed a location planning model for emergency stations of hazardous chemical accidents by considering three factors, location risk, cost, and path. Lutter et al. [22] established a mixed integer linear programming model and used a robust optimization method to analyze the location problem of emergency facilities. There are also scholars who consider the above problems together. For example, Abounacer et al. [23] designed a multi-objective optimization model for location and path planning problems in the emergency response phase and proposed the corresponding exact algorithm. Li et al. [24] developed a two-layer planning model for the integrated optimization of work scheduling and logistics optimization problems in the post-disaster emergency response phase. Geng et al. [25] constructed an emergency relief warehouse site selection and relief material distribution model to reduce the pain perception cost of affected people. Ghasemi et al. [26] combined stochastic planning and game theory approaches to construct models for emergency facility location, humanitarian relief path optimization, and emergency material inventory.

In the emergency response phase, the uncertainty of disaster evolution is high. Some scholars have further studied how to efficiently carry out rescue activities after a disaster incident. Wex et al. [27] studied the allocation and scheduling of emergency rescue units in the context of natural disasters. On this basis, different scholars considered different objective functions and constructed corresponding emergency rescue scheduling models. Tirkolaee et al. [28] developed an allocation and scheduling model for emergency rescue units with learning effects to minimize rescue time and delay. Nayeri et al. [29] designed a robust optimization model to minimize rescue time and verified the model's effectiveness in flooding cases. Emergency rescue in a chemical park during a flooding disaster consists of a series of activities with resource and timing relationship constraints from start to end, and the whole rescue process has prominent project-based and "one-time" characteristics. A project is defined as a temporary effort that is undertaken to create a unique product or service [30]. According to the definition of "project", we can find that the project has the characteristic of "one-time". Therefore, this paper views emergency rescue work as an emergency project scheduling problem and uses project scheduling theory to optimize the flood emergency rescue process. The service created by the rescue project is the minimization of damage to the affected people and the deliverable of the rescue project is the successful completion of the rescue work. The multi-mode resource-constrained project scheduling problem (MRCPSP) [31,32] is mainly used to provide a baseline plan for a project under the condition that resource constraints and timing relationships are satisfied. This plan specifies the start time of activities in the project and allocates the required resources to those activities. In addition, multiple execution modes exist for activities in the project. The MRCPSP can be further divided into two cases, proactive scheduling and reactive scheduling, where proactive scheduling is used to determine the initial scheduling plan and guide project execution [33]. Reactive scheduling is used to dynamically adjust the initial scheduling plan according to the actual situation to ensure the achievement of project goals [34].

The resource-constrained project scheduling problem under various uncertainties has been studied. Chakrabortty et al. [35] investigated an event-based reactive project scheduling approach under resource disruption. Van de Vonder et al. [34] designed a reactive scheduling model for the baseline planning of rehabilitation projects in response to the uncertainty of the activity duration. Ma et al. [36] analyzed the impact of two types of uncertain environments, resource and activity duration, on the robustness of proactive project scheduling. Lambrechts et al. [37] designed an integrated proactive–reactive project scheduling strategy by considering the uncertainty of resource volume. Some scholars have already studied the project scheduling problem in the context of emergency response. Yan et al. [38] first combined project scheduling theory to optimize rescue activities in order to minimize rescue project duration for maritime disasters. To address the problem of a single-mode emergency rescue project, Wang et al. [39] integrated and optimized proactive and reactive project scheduling to maximize rescue activity robustness and minimize

adjustment losses. The similarities and differences between this paper and the above references are summarized in Table 1. Table 1 compares the following four main aspects: uncertain factors, objective functions, activity execution modes, and scheduling methods.

**Table 1.** Comparison of the main research works related to this study.

| References | | [27] | [28] | [35] | [36] | [37] | [38] | [39] | [40] | [41] | This Paper |
|---|---|---|---|---|---|---|---|---|---|---|---|
| Uncertain factors | Resources | | | √ | √ | √ | √ | √ | √ | √ | √ |
| | Activity duration | | | | √ | | | √ | | | √ |
| | Uncertain scenarios | √ | √ | | | | | | √ | | √ |
| Objective functions | Loss of affected people | | | | | | | | √ | | √ |
| | Robustness | | | √ | √ | √ | √ | | √ | | √ |
| | Cost | | | | | | √ | | | | |
| | Rescue efficiency | √ | √ | | | | √ | √ | | √ | √ |
| Execution Modes | Single mode | √ | √ | √ | √ | √ | √ | √ | √ | | |
| | Multi-mode | | | | | | | | | √ | √ |
| Scheduling methods | Proactive scheduling | √ | √ | | √ | | √ | | √ | | |
| | Reactive scheduling | | | √ | | | | | | √ | |
| | Proactive–reactive integrated scheduling | | | | | √ | | √ | | | √ |

By analyzing the above literature and comparing the information in Table 1, we find that current scholars have made scientific and effective assessments of the safety risks, accident hazards, and emergency response capabilities of chemical parks under floods before the occurrence of disasters and accidents. In the process of disasters and accidents, some scholars have done sufficient research on emergency path planning, emergency resource scheduling, and emergency facility location. In addition, some scholars have also conducted research on how to carry out rescue activities efficiently. However, the current research has the following three shortcomings. First, emergency rescue is a matter of system engineering. There are many studies on capacity assessment, location of emergency facilities, path planning, and material scheduling for emergency rescue in chemical parks after flooding. However, there is still room for research and development on how to scientifically and reasonably formulate and implement rescue plans when rescue resources arrive at the affected sites. By reviewing the literature, we found that the characteristics of emergency rescue and project scheduling are consistent, and project scheduling theory is applicable to the study of emergency rescue scheduling. However, most of the previous studies on project scheduling are based on industrial projects, which are rarely applied to the scenario of flooding emergency rescue in chemical parks. Flood emergency rescue in chemical parks has unique characteristics, such as being time-sensitive and having a dynamic and weak economy, which are studied in this paper. Second, with respect to emergency rescue scheduling, the current research on emergency rescue project scheduling problems either considers only single-mode proactive–reactive scheduling or only multi-mode reactive scheduling. There are no emergency rescue studies that combine multi-mode and proactive–reactive scheduling. Third, most of the existing emergency relief project scheduling studies are aimed at maximizing the robustness of the project plan or minimizing the project duration, without considering reducing the loss of affected people as the primary goal. In order to optimize and improve the above deficiencies, taking a chemical park suffering from flooding as a background and considering the dynamic evolution of disaster situations and the phase update of emergency resources, this paper combines project scheduling theory to construct a multi-mode emergency rescue proactive–reactive integrated scheduling model in order to coordinate limited emergency resources to improve the efficiency of rescue decisions, minimize the loss of affected people, and provide theoretical support for sustainable emergency rescue operations under flooding disasters.

The rest of this paper is organized as follows: Section 2 defines the parameters and variables, and constructs a proactive–reactive multi-mode emergency rescue integrated scheduling model. Section 3 designs the ant colony algorithm to solve the model constructed in Section 2. Section 4 introduces a rescue case of a chemical industry park suffering from a flood disaster in Yunnan Province to verify the effectiveness of the model and algorithm and analyzes the calculation results. Section 5 summarizes the findings and suggests future research directions.

## 2. Problem Definition

### 2.1. Problem Description and Explanation of Parameters

#### 2.1.1. Problem Description

Emergency rescue in a chemical park during a flooding disaster consists of a series of activities with resource and timing relationship constraints from start to end. In this paper, the whole rescue process is considered as a project, and the project scheduling theory is applied to the whole emergency rescue process. Heavy rains cause flooding in a chemical park. The emergency operations command center (EOCC) arranges an emergency rescue scheduling baseline plan based on the affected situation, as well as on-site data to provide guidance for the preparation and implementation of emergency rescue. The establishment of the emergency rescue scheduling baseline plan belongs to the category of proactive scheduling. With the dynamic evolution of the disaster situation and the dynamic update of emergency resources, at some point the benchmark plan cannot meet the rescue needs, and the EOCC will adjust the rescue plan based on the latest disaster data information. Adjustments to the baseline plan fall under the category of reactive scheduling. The two scheduling models are correlated through the emergency rescue baseline plan to form proactive–reactive integrated scheduling, which achieves the purpose of reducing the loss of affected people and promoting sustainable development.

The integrated proactive–reactive emergency rescue scheduling process is shown in Figure 1.

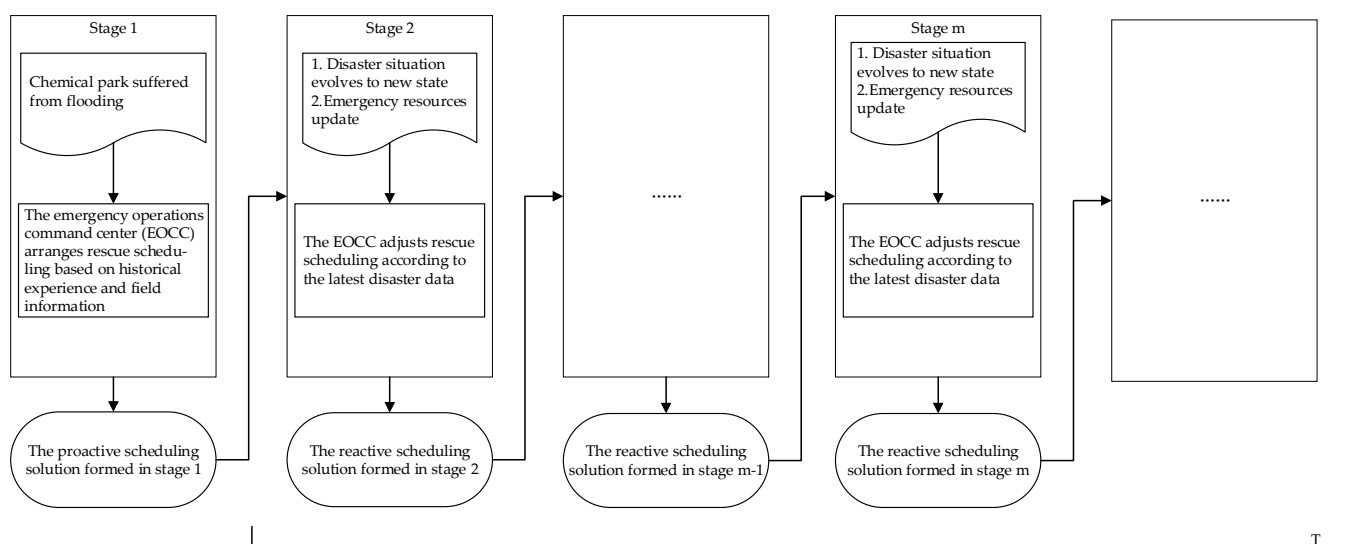

**Figure 1.** The integrated proactive-reactive emergency rescue scheduling process.

#### 2.1.2. Explanations of Parameters and Variables

The multi-mode emergency rescue project of the chemical park under flooding is represented by an $AoN(Activity - on - Node)$ network diagram $G = (N, A)$, where $N$ denotes the set of rescue activity nodes and $A$ denotes the set of logical relationships between each rescue activity. The set $N$ consists of $n + 2$ activities, where activity 0 and activity $n + 1$ are set as dummy activities, and they denote the start and end of the emergency rescue project,

respectively. Assume that the kind of resources required for non-dummy rescue activity $i(i = 1, 2, \ldots, n)$ in the project is $K$ and the capacity of the $kth(k = 1, 2, \ldots, K)$ resource is $R_k$. There are $M_i$ execution modes for activity $i(i = 1, 2, \ldots, n)$. When taking different modes $m(m = 1, 2, \ldots, M_i)$ of execution, the activity $i$ has a different duration $p_{im}$ and consumes a different amount of the $kth$ resource $r_{imk}$. The resources occupied during the execution will be released after the activity is executed. Assume that the rescue duration of activity $i$ is a normally distributed random variable with mean $E(p_{im})$ and standard deviation $\sigma(p_{im})$. $P_i$ denotes the set of immediately preceding activity $i$, $A_t$ denotes the set of activities being executed at moment $t$, and $T$ denotes the upper limit of the rescue project duration. Assuming that the start time of the rescue activity $i$ is $s_i$ and the end time is $f_i$, the baseline plan generated by the proactive emergency rescue scheduling is $S^b = \left( s_0^b, s_1^b, \ldots, s_{n+1}^b \right)$, and the corresponding set of execution modes is $X^b = \left( x_{0m}^b, x_{1m}^b, \ldots, x_{n+1,m}^b \right)$. If the activity $i$ takes the execution mode $m$, then $x_{im}^b = 1$; otherwise, $x_{im}^b = 0$.

The detailed parameters and variables of the emergency rescue project scheduling mathematical model are as follows:

| | |
|---|---|
| $i$ | Activity number, $i = 0, 1, 2, \ldots, n+1$, where $n$ indicates the total number of real activities included in the rescue project. |
| $t$ | Time period number, $t = 0, 1, 2, \ldots, T$, where $T$ indicates the maximum project duration. |
| $k$ | Resource serial number $k = 0, 1, 2, \ldots, K$, where $K$ indicates the number of updateable resource species in the project. |
| $p_{im}$ | Duration when activity $i$ takes mode $m$ execution. |
| $d_i$ | Deadline for event $i$. |
| $P_i$ | The set of activities immediately preceding activity $i$. |
| $s_i$ | Start time of the activity $i$. |
| $F_t$ | The set of completed activities at moment $t$. |
| $A_t$ | The set of activities being executed at moment $t$. |
| $U_t$ | The set of activities that have not yet started at moment $t$. |
| $R_k$ | Capacity of the $kth$ updateable resource. |
| $M_i$ | Activity $i$ has $M_i$ execution modes. |
| $r_{imk}$ | The demand for resource $k$ when activity $i$ adopts mode $m$. |

*2.2. Model Construction*

2.2.1. Construction of a Proactive Emergency Rescue Scheduling Model

In proactive emergency rescue project scheduling, $\forall (i, j) \in A$, the start time of activity $i$ is recorded as $s_i^b$. In the initial stage of emergency rescue, we should first adhere to the orientation of life and the primary goal is to rescue the affected people and reduce their losses [42]. Drawing on the meaning of the loss function of the affected people [40], the marginal suffering of the affected people will remain constant or even increase with an increase in the duration of rescue activities. Considering the different disaster levels, it is assumed that the disaster level is represented by $\theta$. A larger value of $\theta$ represents a higher severity of the disaster, which also means greater damage to the affected people. In addition, there are factors, such as the extent to which material needs are met, that also affect the loss of people affected. To simplify the problem, all other influencing factors are represented by the effects of the duration of rescue activities. Any increase in the duration of rescue activities will increase the suffering or loss of the affected people. In this paper, we use the suffering of the affected people to represent the loss of the affected people and, according to the meaning of the loss function of the affected people [40], the function of the loss of the affected people and the duration of the rescue activity is a convex function. The total duration of rescue activity $i$ is denoted as $\Delta_i$. Its calculation formula is as follows:

$$\Delta_i = \min \left( s_j^b \right) - s_i^b, \forall (i, j) \in A, i = 0, 1, 2, \ldots, n$$

The duration of each rescue activity is different, and the standard deviation of the duration also differs, so the degree of suffering and loss caused by different rescue activities

to the affected people is not the same; based on this, it is necessary to borrow the activity weight coefficient $\omega_i$ to weigh them. The activity weight coefficient $\omega_i$ is defined as follows:

$$\omega_i = \frac{\sigma_i}{\sum_{j=0}^{n} \sigma_j}$$

According to the formula, the value of the activity weight coefficient is mainly determined according to the standard deviation of rescue activities. The larger the $\omega_i$ value, the higher the variability of activity $i$, and the greater its impact on the rescue plan. Based on the above information, a proactive emergency response project scheduling model is established, as shown below:

$$minz_1 = \sum_{i=0}^{n+1} \omega_i (\Delta_i)^\theta \tag{1}$$

*s.t.*

$$s_0^b = 0 \tag{2}$$

$$s_i^b + \sum_{m=1}^{M_i} E(p_{im}) x_{im} \leq s_j^b, \forall (i,j) \in A, i \in N, j \in N \tag{3}$$

$$s_{n+1}^b \leq T \tag{4}$$

$$\sum_{i \in A_t} \sum_{m=1}^{M} r_{imk} x_{im} \leq R_k, k = 1, 2, \ldots, K \tag{5}$$

$$\sum_{m=1}^{M} x_{im} = 1, i = 0, 1, 2, \ldots, n+1 \tag{6}$$

$$x_{im} = \begin{cases} 1, & \text{If the rescue activity i adopts mode m} \\ 0, & \text{otherwise} \end{cases} \tag{7}$$

$$s_i^b \in N^0 \tag{8}$$

Equation (1) is the objective function of the proactive scheduling model, which indicates minimizing the total loss of affected people. Equation (2) represents that the start time of the rescue project is moment 0, i.e., the start time of the dummy activity 0 is moment 0. Equation (3) represents the immediately preceding relationship constraint among the rescue activities. Equation (4) represents the end time of the rescue project, i.e., the start time of the dummy activity $n+1$ does not exceed the upper limit of the rescue duration $T$. Equation (5) represents the resource constraint. Equation (6) shows that only one mode can be selected for each rescue activity when it is executed. Equation (7) represents the definition domain of the execution mode decision variable. Equation (8) represents that the start time of each rescue activity is non-negative.

### 2.2.2. Construction of a Reactive Emergency Rescue Scheduling Model

Most current reactive scheduling problems have the goal of maximizing scheduling stability [34,39], or the goal is to minimize the cost of reactive scheduling [43,44]. Combined with the characteristics of flooding emergency rescue in chemical parks, there is a phased renewal of emergency resources, which leads to the acceleration of rescue progress. In other words, the rescue scheduling plan can be better. Most previous studies have focused on making the deviation between the old and new scheduling solutions as small as possible but cannot guarantee that the new scheduling solution is optimal in the new emergency rescue environment [41]. Based on this, an improved reactive emergency rescue scheduling model is constructed in this paper to solve the above problems.

During the execution of the rescue project, it is assumed that the baseline plan for the rescue needs to be reactively adjusted due to the occurrence of disruptive events such as the evolution of the disaster situation or the dynamic update of emergency resources at

moment $q$. In the new emergency rescue environment at moment $q$, the shortest rescue duration that can be achieved is $T^q$. The set of completed rescue activities at this time are denoted as $F_q$, the set of activities being performed are denoted as $A_q$, and the set of activities that have not yet been implemented are denoted as $U_q$. Let the rescue solution generated by reactive scheduling be $S^q = \left( s_0^q, s_1^q, \ldots, s_{n+1}^q \right)$ and the corresponding set of execution modes be $X^q = \left( x_{0m}^q, x_{1m}^q, \ldots, x_{n+1,m}^q \right)$. If activity $i$ takes the execution mode $m$, then $x_{im}^q = 1$; otherwise, $x_{im}^q = 0$.

Based on the above information, a reactive emergency response project scheduling model is established, as shown below:

$$minz_2 = \sum_{i=0}^{n+1} \left| s_i^q - s_i^b \right| + \sum_{i=0}^{n+1} \left| f_i^q - f_i^b \right| \tag{9}$$

*s.t.*

$$s_{n+1}^q = T^q \tag{10}$$

$$s_i^q = s_i^b, i \in F_q \cup A_q \tag{11}$$

$$x_{im}^q = x_{im}^b, i \in F_q \cup A_q \tag{12}$$

$$f_i^q = f_i^b, i \in F_q \cup A_q \tag{13}$$

$$s_i^q + \sum_{m=1}^{M_i} E(p_{im}) x_{im}^q = f_i^q \tag{14}$$

$$s_i^q + \sum_{m=1}^{M_i} E(p_{im}) x_{im}^q \leq s_j^q, \forall (i,j) \in A, i \in N, j \in U_q \tag{15}$$

$$s_i^q \in N^0 \tag{16}$$

Equation (9) is the objective function of the reactive scheduling model. The function consists of two components, which represent the total deviation of the activity start time and end time before and after the reactive scheduling, respectively. The sum of the two is minimized in order to control the minimum change in the start time of the activity and the selected execution mode when adjustments are made to the baseline rescue scheduling plan. Equation (10) shows that the total duration of the reactive rescue scheduling plan generated at moment of adjustment $q$ is equal to the shortest rescue duration that can be achieved at moment $q$. This constraint ensures that the new reactive scheduling plan has the shortest duration. Equations (11)–(13) represent the rescue activities that have been completed or are being executed at moment $q$ that have the same start time, execution mode, and end time as the baseline rescue plan. Equation (14) indicates the end time of the activity. Equation (15) indicates the immediately preceding relationship constraint between each rescue activity at moment $q$. Equation (16) shows that the start time of each rescue activity at moment $q$ is non-negative. In addition, Equations (5)–(7) of the proactive emergency rescue project scheduling model are still applicable in the reactive scheduling model.

## 3. Solution Algorithm

It has been shown that MRCPSP is an NP-hard problem [45], and the heuristic algorithm is an effective algorithm for solving this problem. As a mature heuristic algorithm, the ant colony algorithm is suitable for solving this problem due to its strong robustness and easy parallel computation. Therefore, this paper draws on the idea of the ant colony algorithm [46–48] and improves it to solve the proposed multi-mode emergency rescue proactive–reactive integrated scheduling problem.

In this paper, the one-dimensional real numbers based on the execution mode of rescue activities are encoded, and the activity network graph is constructed through this encoding plan. A roulette wheel algorithm based on pheromone terms and heuristic terms

is then implemented for state transfer. Under the constraints of emergency resources and temporal relations, the process of ants crawling away on the network graph according to the state transfer strategy is the solution to this problem. The flow of the application of the ant colony algorithm is shown in Figure 2. The ant colony algorithm in this paper is divided into five main steps, which are reading the dataset and initializing the parameters, developing a state transfer strategy, updating the pheromone, developing activity priority rules, and updating the activity list. Sections 3.1–3.5 describe the details of the algorithm.

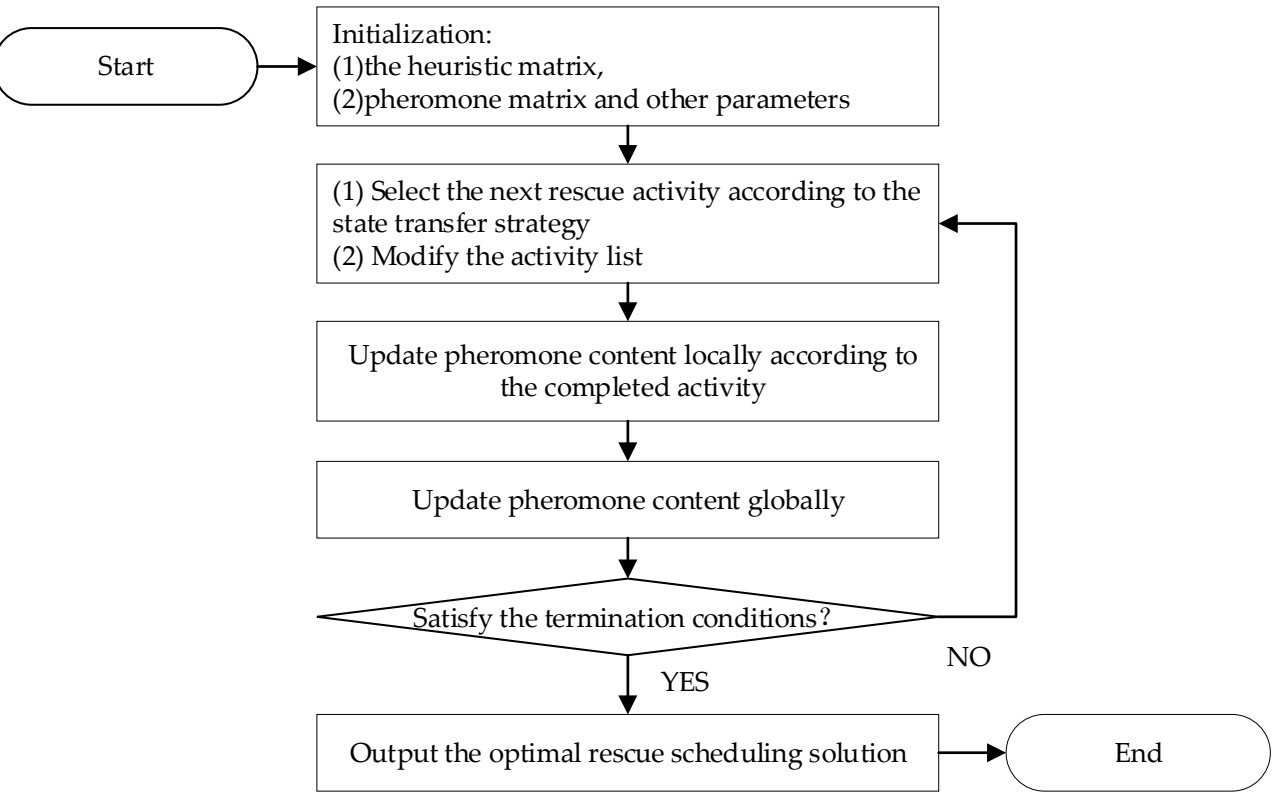

**Figure 2.** Flow chart of the application of ant colony algorithm.

### 3.1. Initialization

3.1.1. Initialization of the Heuristic Term Matrix

The heuristic term matrix is used to record the possibility of selecting each rescue activity, and its value is related to the selected greedy strategy. In this paper, the ratio of the number of emergency resources required for a rescue activity to the total number of resources is used as the greedy strategy for activity selection. Assume that $\eta_{ij}(\mathrm{m})$ denotes the heuristic function for transferring activity $i$ to activity $j$ under mode $m$. Its formula is as follows:

$$\eta_{ij}(\mathrm{m}) = \frac{\sum_{k \in K} r_{imk}}{\sum_{k \in K} R_k}, m \in M_i \tag{17}$$

3.1.2. Initialization of the Pheromone Matrix

The pheromone content matrix records the values of pheromones left by ants on each path [46]. $Q$ is the quality factor, which represents the total amount of pheromones left behind by ants after walking one path, and $N_A$ denotes the total number of rescue activities passed by ant $a$ after walking one path. The strategy used to initialize the pheromone

content $\tau_{ij}(m)$ of each rescue activity $i$ is the ratio of the quality factor $Q$ to the number of activities $N_A$, which is calculated as follows:

$$\tau_{ij}(m) = \frac{Q}{N_A} \tag{18}$$

*3.2. State Transfer Strategy*

The state transfer strategy is the method taken by ants to choose the next rescue activity in the candidate activity list, and this paper mainly adopts the roulette strategy. The ant selects the probability of the next rescue activity by considering both the strength of the heuristic item information and pheromones in the current activity. Assume that the state transfer probability of ant $a$ from activity $i$ to activity $j$ at the moment $t$ is $P_{ij}^a(t, m)$, which is calculated by the following formula:

$$P_{ij}^a(t, m) = \begin{cases} \dfrac{[\tau_{ij}(t, m)]^{\alpha} [\eta_{ij}(t, m)]^{\beta}}{\sum_{s \in E_a} [\tau_{is}(t, m)]^{\alpha} [\eta_{is}(t, m)]^{\beta}}, & j \in E_a \\ 0, & otherwise \end{cases} \tag{19}$$

where $E_a$ denotes the set of optional rescue activities for ant $a$ in the next step. Parameter $\alpha$ denotes the relative importance of the heuristic information in the ant's path selection during its movement, and parameter $\beta$ denotes the relative importance of the pheromone in the path selection of the ant during its movement.

A roulette machine is created according to the probability of each activity. A number from 0 to 1 is then randomly generated, and the rescue activity on the roulette machine corresponding to this number is selected. This method ensures that the rescue activities with high pheromone intensity are selected in preference and also increases the randomness of the ant search.

*3.3. Pheromone Update*

The pheromone update is divided into two parts. The first part is the local update pheromone $\tau_{ij}^l$. The pheromone is updated each time the ant selects an activity and its mode. The calculation formula is as follows:

$$\tau_{ij}^l(t + 1, m) = (1 - \rho) \times \tau_{ij}(t, m) + \rho \times 1/\Delta_i \tag{20}$$

The second part is the global update pheromone $\tau_{ij}^g$. After all ants have walked, the one that spends the shortest time is selected to significantly enhance the pheromone content on the path it has walked, making the next ants more likely to choose that path. The calculation formula is as follows:

$$\tau_{ij}^g(t + 1, m) = (1 - \rho) \times \tau_{ij}(t, m) + \rho \times 1/T \tag{21}$$

where $T$ is the completion duration of the scheduling plan obtained by the ant in this cycle. The pheromone volatility factor $\rho$ determines the pheromone retention time on the path [46]. To prevent falling into a local optimum, the value of $\rho$ is dynamically adjusted in stages as the number of iterations $n$ increases. It provides support for the search ability and computational efficiency of the algorithm. Its formula is as follows:

$$\rho = \begin{cases} 0.1, & 0 \leq n < 0.25N \\ 0.2, & 0.25N \leq n < 0.75N \\ 0.5, & 0.75N \leq n < N \end{cases} \tag{22}$$

### 3.4. Establish Activity Priority Rules

Under the premise of satisfying the resource constraints and timing constraints among rescue activities, the minimum start time to complete an activity is used as the criterion for prioritizing the activities.

### 3.5. Update the Event List

The activity waiting list update method includes two parts. The first part is removing an activity from the activity list after it is executed. The second part is to judge all the immediate preceding events of an activity after its subsequent events are executed, and if all of them are executed, the subsequent activities are added to the activity waiting list according to the logical relationship constraint.

### 3.6. Solution of Reactive Emergency Rescue Scheduling Model

The algorithmic process in Sections 3.1–3.5 is designed mainly for the proactive emergency scheduling model, and the algorithm can be applied to the solution of the reactive scheduling model with minor adjustments. The specific adjustments are as follows: the objective function is changed from $minz_1$ to $minz_2$, and the activity waiting list is adjusted to apply only to the activities in the set of activities $U_q$ that have not yet been executed.

The above ant colony algorithm is programmed in Python and was run on a personal computer with a CPU main frequency of 1.6 GHz and 8 GB of memory.

## 4. Computational Experiments

### 4.1. Case Data Description

In order to verify the effectiveness of the model and algorithm, this paper uses the emergency rescue process after a flooding disaster in a chemical park in Yunnan Province as an example. We use the actual rescue theoretical data obtained by scientific analysis and expert interviews as a reference for simulation verification [49,50].

Figure 3 shows the logical relationship between each rescue activity in the form of a network diagram. The rescue process of this chemical park can be divided into the three following stages: on-site disaster assessment and warning, emergency rescue, and post-disaster resilience recovery. The emergency rescue stage can be divided into the four following parts: disaster victim relief, park asset protection, park pollution treatment, and public facility repair according to the different rescue objects. In this paper, we focus on the rescue process after emergency resources arrive at the disaster site and divide the rescue process into 26 activities based on the actual rescue situation. For the completeness of the rescue network structure, this paper sets two dummy activities, 0 and 27, and assumes that the disaster level $\theta = 2$.

In this paper, emergency rescue resources are grouped into two categories: rescue personnel ($k = 1$) and rescue equipment ($k = 2$). We assume that the total amount of the two rescue resources are $(R_1, R_2)$. The expected rescue resources available for each rescue activity $i$ under the different execution mode $m$ are $(r_{im1}, r_{im2})$, where $r_{im1}$ represents the number of different types of rescue personnel required for rescue activity $i$ in execution mode $m$, and $r_{im2}$ represents the number of rescue equipment required for rescue activity $i$ in execution mode $m$. To ensure the accuracy of the data, the standard deviation of the duration of the rescue activities is determined by consulting experts. The expected duration of rescue activities and the expected input emergency resources in different modes are obtained by scientific analysis based on the actual rescue data, which are shown in Table 2. Table 2 gives the basic data of 26 non-virtual rescue activities in different execution modes, including the expected duration of rescue activities, expected input resources, activity weights, and sequence of events.

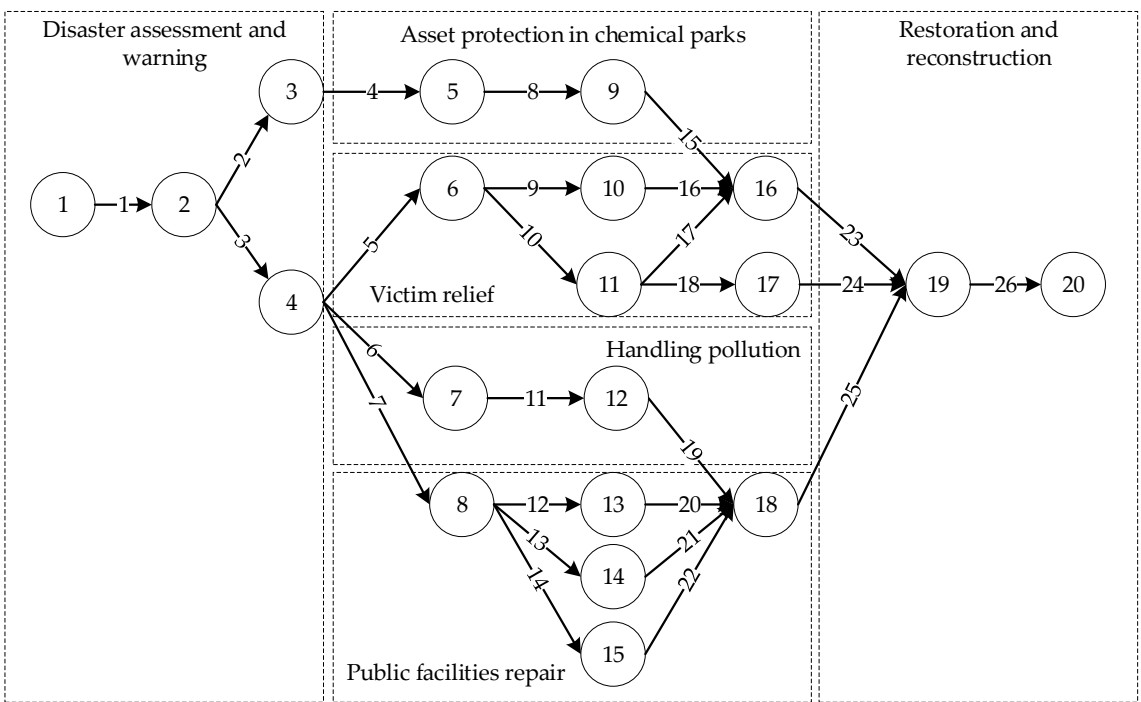

**Figure 3.** Emergency rescue activities logic relationship diagram.

**Table 2.** The basic data of emergency rescue in different modes.

| Activity Number | Activity Name | Normal Mode | | Emergency Mode | | Activity Weights | Sequence of Events |
|---|---|---|---|---|---|---|---|
| | | Expected Duration | Expected Input Resources | Expected Duration | Expected Input Resources | | |
| 0 | Disaster Occurrence | 0 | (0,0) | 0 | (0,0) | 0 | 0–1 |
| 1 | Disaster investigation and assessment | 24 | (60,20) | 19 | (72,24) | 0.01 | 1–2 |
| 2 | Hidden danger inspection | 20 | (80,30) | 16 | (96,36) | 0.01 | 2–3 |
| 3 | Geological disaster prevention | 24 | (90,40) | 16 | (102,45) | 0.04 | 2–4 |
| 4 | Pumping and draining | 68 | (70,25) | 54 | (84,30) | 0.06 | 3–5 |
| 5 | Search and rescue | 48 | (240,70) | 36 | (288,84) | 0.07 | 4–6 |
| 6 | On-site traffic control | 60 | (60,20) | 48 | (72,24) | 0.01 | 4–7 |
| 7 | Fire fighting | 24 | (80,20) | 16 | (96,24) | 0.07 | 4–8 |
| 8 | Road repair | 22 | (120,36) | 16 | (140,40) | 0.06 | 5–9 |
| 9 | Evacuation of residents | 12 | (90,30) | 8 | (110,36) | 0.04 | 6–10 |
| 10 | Transfer of affected people | 6 | (80,40) | 5 | (96,48) | 0.04 | 6–11 |
| 11 | Pollution treatment | 96 | (120,60) | 65 | (142,72) | 0.06 | 7–12 |
| 12 | Electricity repair | 12 | (30,15) | 10 | (36,18) | 0.01 | 8–13 |
| 13 | Gas supply repair | 20 | (20,10) | 15 | (22,12) | 0.03 | 8–14 |
| 14 | Water supply repair | 26 | (20,10) | 18 | (24,12) | 0.03 | 8–15 |
| 15 | Asset transfer protection | 19 | (50,20) | 12 | (58,24) | 0.04 | 9–16 |
| 16 | Communication guarantee | 24 | (16,8) | 18 | (20,10) | 0.03 | 10–16 |
| 17 | Material supply | 48 | (73,20) | 28 | (96,24) | 0.04 | 11–16 |
| 18 | Medical assistance | 36 | (100,30) | 25 | (115,36) | 0.06 | 11–17 |
| 19 | Environmental monitoring | 20 | (20,10) | 20 | (20,10) | 0.01 | 12–18 |
| 20 | Electricity guarantee | 12 | (15,8) | 10 | (18,10) | 0.04 | 13–18 |
| 21 | Gas supply guarantee | 12 | (10,5) | 8 | (12,6) | 0.01 | 14–18 |
| 22 | Water supply guarantee | 12 | (10,5) | 9 | (12,6) | 0.04 | 15–18 |
| 23 | Epidemic prevention | 19 | (50,20) | 12 | (58,24) | 0.01 | 16–19 |
| 24 | Resettlement of people | 30 | (160,80) | 22 | (192,96) | 0.06 | 17–19 |
| 25 | Hidden danger detection | 40 | (60,20) | 32 | (72,24) | 0.04 | 18–19 |
| 26 | Reconstruction and repair | 72 | (180,80) | 58 | (216,96) | 0.06 | 19–20 |
| 27 | End | 0 | (0,0) | 0 | (0,0) | 0 | 20–21 |

### 4.2. Analysis of Calculation Results

The actual rescue completion time of the case (256 h) is taken as the upper limit of the initial rescue time, i.e., $T = 256$ h. The data in Table 2 are brought into the proactive emergency rescue scheduling model. The baseline rescue plan under flooding is calculated as $S^b = (0, 0, 134, 0, 138, 16, 47, 32, 154, 40, 129, 64, 100, 112, 100, 189, 165, 165, 129, 118, 134, 129, 129, 236, 206, 168, 236, 255)$. The corresponding execution model is $X^b = (0, 0, 0, 1, 0, 0, 1, 1, 0, 0, 1, 1, 0, 1, 1, 1, 0, 1, 0, 1, 1, 0, 1, 0, 0, 0, 0, 0)$, and the loss caused to the affected people is 4722.24. $X^b$ is the set of rescue activity execution modes generated by the proactive emergency rescue scheduling model, where 0 means that the rescue activity takes the normal model and 1 means that the rescue activity takes the emergency model. It can be found that $s^b_{27} = 255$ h, which means that the total duration of the baseline plan generated by the proactive scheduling model is 255 h. Compared with the actual rescue completion time, it is shortened by 1 h.

According to the baseline rescue plan $S^b$, the priority activities are search and rescue, evacuation of residents, and other rescue activities for the affected people. This result also reflects the principle of life first rescue.

As emergency rescue activities continue, the disaster situation in the chemical park evolves dynamically and rescue resources arrive at the disaster site one after another, meaning the baseline rescue plan cannot satisfy the current rescue needs. It is necessary to conduct reactive scheduling of the rescue activities based on the baseline rescue plan and the actual situation. Based on the actual situation, three primary reactive schedules are executed during the rescue. At moment 24 of emergency rescue, new emergency resources have arrived at the disaster site, and the total amount of emergency resources is updated from (603,99) to (668,106). At moment 64, emergency resources are replenished again, and the total amount of resources is updated from (668,106) to (735,117). At moment 139, activity 8 is extended by 6 h due to changes in disaster conditions. The specific scheduling results at different moments are shown in Table 3. Column 1 in Table 3 indicates the serial numbers of the three reactive schedules. Column 2 indicates the emergency rescue scheduling plan generated by the three reactive schedules. Column 3 indicates the corresponding rescue duration. Column 4 indicates the loss of affected people from each scheduling. In column 5, the ratio of the reduction in loss of affected people for each response schedule compared to the previous schedule is shown. The data in Table 3 show that after 3 reactive schedules, the rescue duration and the loss of affected people show an overall decreasing trend, in which the rescue duration is shortened by a total of 43 h and the loss of affected people is reduced by 27.90%. These data prove the effectiveness of the model constructed in this paper.

**Table 3.** Reactive rescue scheduling results.

| Serial Number | Reactive Scheduling Solutions | Rescue Duration (h) | Loss of Affected People | Magnitude of Loss Change |
|---|---|---|---|---|
| 1 | (0,0,135,0,141,16,62,24,154,34,129,64,24,42,42,160, 170,155,129,129,129,129,129,129,216,189,155,219,238) | 238 | 3971.83 | −18.89% |
| 2 | (0,0,95,0,115,16,62,24,142,34,64,77,24,42,42,142, 190,142,64,85,85,80,85,205,183,142,205,208) | 208 | 3696.61 | −7.45% |
| 3 | (0,0,95,0,115,16,62,24,145,34,64,77,24,42,42,188, 188,145,64,85,85,80,85,207,177,145,207,212) | 212 | 3639.65 | −1.56% |
| Total | - | - | - | −27.90% |

### 4.3. Sensitivity Analysis

From Table 3, it can be seen that different amounts of emergency resources can have an impact on the duration of rescue activities and the loss of affected people. From the model constructed in Section 2.2, it can be seen that the activity execution mode, the number of emergency resources, the weighting coefficient of rescue activities, and the rescue duration

all have an impact on the emergency rescue work, and a sensitivity analysis is carried out on these key parameters. Assuming that there are four combinations of emergency resources, the specific information is shown in Table 4.

**Table 4.** Different resource portfolios of emergency resources.

| Resource Portfolio | Rescue Personnel | Rescue Equipment |
| --- | --- | --- |
| 1 | 603 | 99 |
| 2 | 608 | 103 |
| 3 | 668 | 106 |
| 4 | 735 | 117 |

Based on the baseline rescue plan and the corresponding solution obtained from the proactive scheduling, the model is solved by varying the total number of emergency resources based on the data in Table 4 while keeping the other parameters constant. The calculation results are shown in Figures 4 and 5.

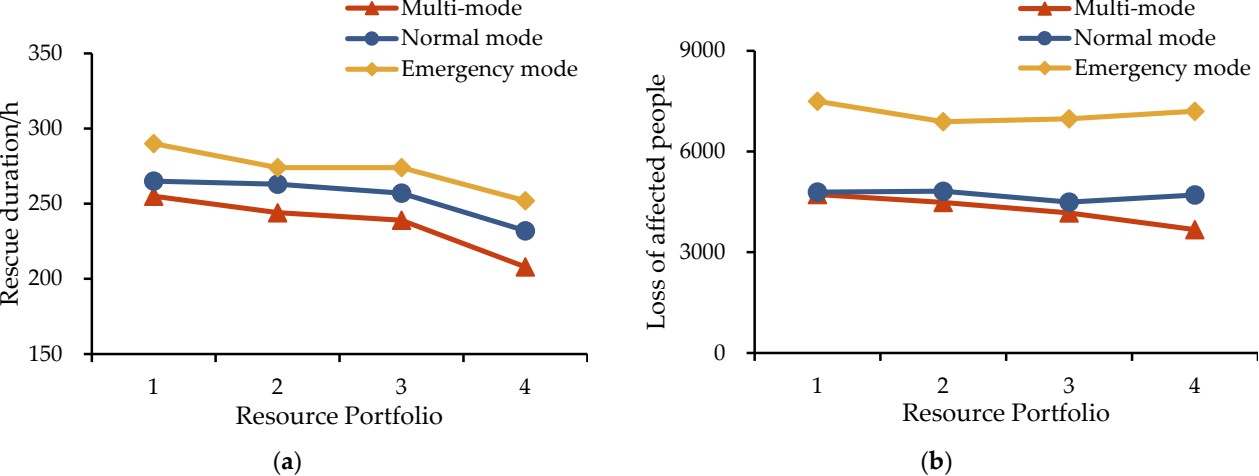

**Figure 4.** (**a**) Rescue duration under different modes. (**b**) The loss of affected people under different modes.

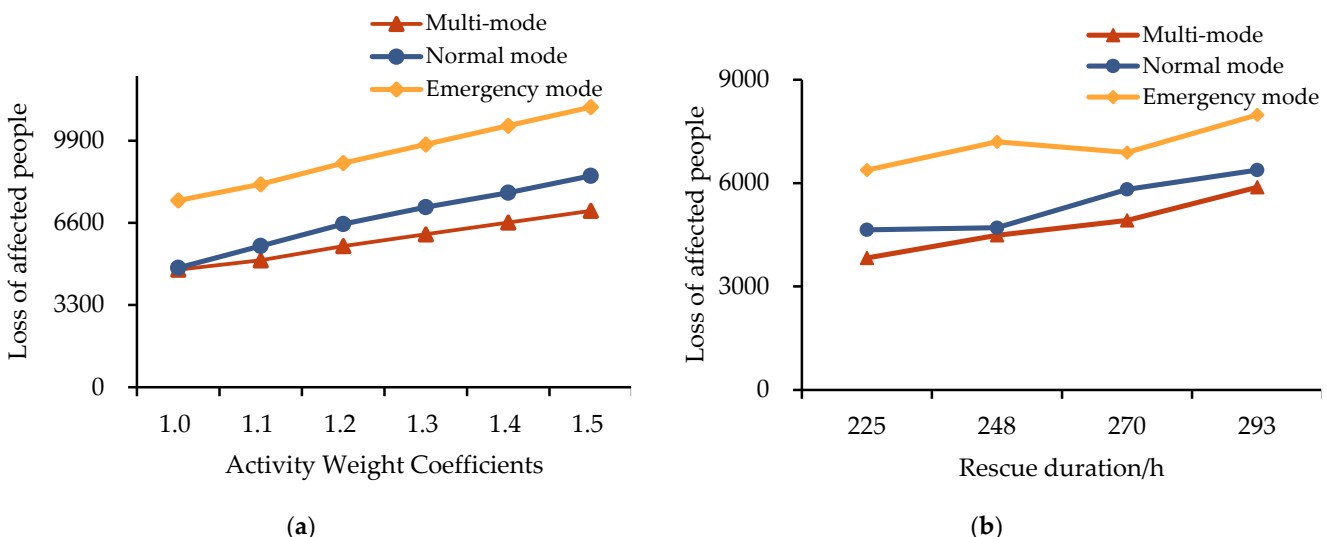

**Figure 5.** (**a**) The loss of affected people under different activity weight coefficient. (**b**) The loss of affected people under different duration of rescue activities.

Figure 4a shows the rescue duration when rescue activities are performed in each of the three execution modes with different combinations of emergency resources. Figure 4b shows the total loss of affected people when rescue activities are performed in each of the three execution modes with different combinations of emergency resources. From Equation (1), it can be seen that the total loss of affected people is not only related to the rescue duration $\Delta_i$ but also to the activity weight coefficient $\omega_i$ and the disaster level $\theta$, and is not linearly related to the rescue duration $\Delta_i$. This leads to the difference between Figure 4a,b.

In Figure 5a, the horizontal coordinates demonstrate the multiplicative relationship of the activity weight coefficients, e.g., 1.3 represents $1.3\omega_i$. The vertical coordinates represent the loss of affected people. Under the three execution modes, the loss of affected people increases with an increase in the activity weight coefficients $\omega_i$. Figure 5b shows an increasing trend in the loss of affected people with an increase in the rescue duration under the three execution modes. In addition, we can also see from Figure 5 that the loss of affected people is always smallest with multi-mode rescue compared to using single-mode rescue.

From the data in Figures 4 and 5, it can be seen that for different combinations of resources, the rescue duration and the loss of affected people are always the largest when taking the emergency mode. The second highest result is obtained when the normal mode is adopted. It is worth noting that the duration of rescue and the loss of affected people are the smallest when multi-mode rescue is adopted. With an increase in emergency resources, the duration of rescue activities and the loss of affected people in multi-mode rescue gradually reduced. All of the above findings demonstrate the effectiveness of adopting multi-mode rescue.

The above findings are further analyzed here. When the rescue activities adopt the normal mode, the duration of each activity increases, resulting in a longer total rescue duration and greater loss of affected people. When the rescue activities adopt the emergency mode, although the duration of each activity is reduced, the total rescue duration and the loss of affected people are not reduced. The reason for this is that in emergency rescue, resources are limited. Although rescue activities in emergency mode have a short duration, the resource demand increases. The limited resources are not able to meet the resource demands of various rescue activities in time. Some emergency resources that cannot be properly arranged may also be idle, resulting in a waste of resources. For these reasons, the overall rescue duration is extended, and losses are increased.

## 5. Conclusions

Carrying out appropriate and effective emergency rescue activities under flooding conditions is essential when building a scientific disaster management system and guaranteeing sustainable development in chemical parks. In this paper, firstly, by reviewing and summarizing the literature, we identified the problem that the current research on the emergency rescue project scheduling problem considers either only single-mode proactive–reactive scheduling or only multi-mode reactive scheduling. In order to solve the above problems, this paper combines project scheduling theory, constructs a proactive–reactive multi-mode emergency rescue integrated scheduling model and optimizes the emergency rescue scheduling process. In addition, an ant colony algorithm was designed to solve the model. By simulating an actual case, the following conclusions were mainly obtained:

1.  Under the constraint of limited resources, the integrated scheduling of multi-mode emergency rescue can improve rescue efficiency and effectively reduce the loss of affected people compared with a single mode.
2.  With an increase in emergency resources, the duration of rescue activities and the loss of affected people are gradually reduced when adopting a multi-modal execution of rescue activities in the new rescue environment.
3.  At the early stages of emergency response, the baseline rescue plan should follow the rescue principle of life first, giving priority to search and rescue, evacuation of residents, and other rescue work for affected people.

The above conclusions can be used as a reference for decision-makers in the emergency rescue process and contribute to the sustainability of emergency rescue activities. In the future, the following issues should be studied further. First, in this paper, there is only one objective function when constructing the proactive emergency rescue scheduling model, minimizing the loss of affected people, and the proactive–reactive emergency rescue collaborative scheduling problem under multiple objectives can be considered in the future to be more realistic, e.g., by considering maximizing robustness and maximizing the satisfaction of affected people's needs. Second, the research on virtual geographic scenes [51] and rapid 3D reproduction [52] is a hot issue in the field of emergency management. Virtual scenes can present rich and clear disaster information, which can significantly improve the level of public disaster perception and can also make the rescue process highly visualized. Virtual scenes can also enhance the understanding of decision makers of the cause, process, and impact of flooding in chemical parks, and improve relief efficiency. In addition, the theory of reinforcement learning is developing rapidly [53]. It is also possible to combine reinforcement learning algorithms to optimize the solution efficiency of the multi-mode emergency rescue integrated scheduling model constructed in this paper, which is more in line with the time-sensitive requirements of emergency rescue.

**Author Contributions:** Conceptualization, B.G. and W.Z.; methodology, B.G.; software, B.G.; validation, B.G.; investigation, B.G. and W.Z.; data curation, B.G. and W.Z.; writing—original draft preparation, B.G.; writing—review and editing, B.G. and W.Z.; supervision, W.Z. All authors have read and agreed to the published version of the manuscript.

**Funding:** This research was funded by the research project of Jianghai Joint Laboratory of Intelligent Security & Emergency Management at University of Chinese Academy of Sciences, grant number E242980401.

**Institutional Review Board Statement:** Not applicable.

**Informed Consent Statement:** Not applicable.

**Data Availability Statement:** The data presented in this study are available on request from the corresponding author.

**Conflicts of Interest:** The authors declare no conflict of interest.

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
