# Peer review of "Research on Integrated Scheduling of Multi-Mode Emergency Rescue for Flooding in Chemical Parks"

_sustainability, doi:10.3390/su15042930_

Round 1

Reviewer 1 Report

The authors clearly presented their model for rescue project scheduling for disaster emergency, and one test case was used to demostrate the model. The paper is overall of high quality, and its language is easy to follow.

A few comments and questions need to be justified or addressed before publication:

1. In the test case of this work, only the rescue resources are updated with the "proactive-reactive" scheduling model. However, the authors stated that the emergency situation might drastically develop, especially with flooding. How does the model leverage such scenario? How such changes are reflected in the model?

2. The proposed model defined "the total loss of affected people" as the objective function to minimize. The function contains an activity weight coefficient as adjustable parameter. How are the values of these parameters determined? It seems very arbitrary as long as they can fit the target in the test case.

3. The definition of "the total loss of affected people" is only depending on duration of rescue activities and disaster level. Is it possible that some loss is not only dependent of these two factors? How does the model handle these kind of losses?

4. In section 3.5, the first two sentences repeated?

5. In the flow chart presented in Figure 2, why the ant colony algorithm is not reflecting the proposed model with "proactive-reactive" feature to consider situation developement?

6. The test case in section 4 referred to "scientific analysis" of a flooding disaster. Proper reference needs to be provided here.

Reviewer 2 Report

The comments are attached as a file called "Reviewer statement for the manuscript". 

Reviewer 3 Report

In the article titled "Research on integrated scheduling of multi-mode emergency rescue for flooding in chemical parks", a proactive-reactive multi-mode emergency rescue integrated scheduling model be constructed to minimize the loss of affected people in the early stage and minimize the sum of the total deviation of the start time and end time of activities before and after reactive scheduling in the later stage.

The original side of the article is indicated by the following sentences. " As the scale of the chemical park industry continues to expand, the impact of flooding on the park's people and surrounding environment increases. Firstly, in order to alleviate the suffering of the affected people, promote the sustainable development of society and the environment, and take into account the characteristics of the dynamic evolution of flooding in chemical parks and the periodic renewal of emergency resources, this paper uses the project scheduling theory to optimize the emergency rescue process. It constructs a proactive-reactive multi-mode emergency rescue integrated scheduling model intending to minimize the loss of affected people in the early stage and minimize the sum of the total deviation of the start time and end time of activities before and after reactive scheduling in the later stage. Secondly, an ant colony algorithm was designed to solve the constructed model. Finally, the effectiveness of the proposed model and solution algorithm is verified by simulations with actual cases. The calculation results show that using proactive-reactive integrated scheduling can improve the efficiency of emergency rescue and reduce the loss of affected people. Moreover, if a multi-mode rescue strategy is adopted, emergency rescue scheduling under four different resource combinations can reduce the rescue duration and the loss of affected people. The model can provide a decision reference for sustainable emergency rescue scheduling in chemical parks under flooding."

According to this reviewer's opinion the originality of the study is not sufficient because of the following issues.

1. The contribution to the paper to the literature is not clear. There have been many studies in the literature on the subject. In the introduction, the authors should clearly state what their contribution in a precise way with respect to previous works. Considering the methods and data used in the study, it is not clear what is new.

2. The authors should give the previous works in the study about the subject, and then make a comparison to indicate the novelty of the study.

3. The authors should explain the specific hazards of flooding disasters to chemical parks and the relevant factors involved in the accident.

4. The units of the objective function and its meaning should be stated. For example, the value in “The corresponding execution model is Xb = (0, 0, 0, 1, 0, 0, 1, 1, 0, 0, 1, 1, 0, 1, 1, 1, 0, 1, 0, 1, 1, 0, 1, 0, 0, 0, 0, 0), and the loss of affected people is 4722.24”.

5. In 4.3 Sensitivity Analysis, fewer data analysis dimensions are not sufficient to illustrate the usefulness of the method. In another words, the obtained results need considerably more explanation.

6. What is the reason of the difference between Figure 4(a) and Figure 4(b). Can it be analyzed?

7. The language needs to be strengthened.

Round 2

Reviewer 2 Report

Thank you very much for your interest in the reviewer comments.  

Author Response

We feel great thanks for your professional review work on our article. 

Reviewer 3 Report

After revision, the revised paper is satisfactory. The following points are for further improvement by the author:

1. The language needs to be strengthened except for grammar.

2. The contrast between the text and the cited literature is not enough to reflect the innovation of this paper.Table may be a good idea to indicates innovation, where you can list the similarities and differences between this article and the literature in the table.

Author Response

We feel great thanks for your professional review work on our article. Please see the attachment for details of the revisions.
